# Socio-Demographic Factors Involved in a Low-Incidence Phase of SARS-CoV-2 Spread in Sicily, Italy

**DOI:** 10.3390/healthcare9070867

**Published:** 2021-07-09

**Authors:** Emanuele Amodio, Michele Battisti, Carmelo Massimo Maida, Maurizio Zarcone, Alessandra Casuccio, Francesco Vitale

**Affiliations:** 1Department of Health, Promotion, Mother and Child Care, Internal Medicine and Medical Specialties, University of Palermo, Piazza delle Cliniche n. 2, 90127 Palermo, Italy; carmelo.maida@unipa.it (C.M.M.); maurizio.zarcone@policlinico.pa.it (M.Z.); alessandra.casuccio@unipa.it (A.C.); francesco.vitale@unipa.it (F.V.); 2Department of Law, University of Palermo, Via Maqueda n. 172, 90134 Palermo, Italy; michele.battisti@unipa.it

**Keywords:** SARS-CoV-2, outcomes, socio-demographic factors, census section

## Abstract

Background: The present study analysed SARS-CoV-2 cases observed in Sicily and investigated social determinants that could have an impact on the virus spread. Methods: SARS-CoV-2 cases observed among Sicilian residents between the 1 February 2020 and 15 October 2020 have been included in the analyses. Age, sex, date of infection detection, residency, clinical outcomes, and exposure route have been evaluated. Each case has been linked to the census section of residency and its socio-demographic data. Results: A total of 10,114 patients (202.3 cases per 100,000 residents; 95% CI = 198.4–206.2) were analysed: 45.4% were asymptomatic and 3.62% were deceased during follow-up. Asymptomatic or mild cases were more frequent among young groups. A multivariable analysis found a reduced risk of SARS-CoV-2 cases was found in census sections with higher male prevalence (adj-OR = 0.99, 95% CI = 0.99–0.99; *p* < 0.001) and presence of immigrants (adj-OR = 0.89, 95% CI 0.86–0.92; *p* < 0.001). Proportion of residents aged <15 years, residents with a university degree, residents with secondary education, extra-urban mobility, presence of home for rent, and presence of more than five homes per building were found to increase the risk of SARS-CoV-2 incidence. Conclusion: Routinely collected socio-demographic data can be predictors of SARS-CoV-2 risk infection and they may have a role in mapping high risk micro-areas for virus transmission.

## 1. Introduction

The emergence of the novel severe acute respiratory syndrome coronavirus 2 (SARS-CoV-2) has significantly changed the lives of a large part of the world population [1]. In just a few months, the virus has spread worldwide and, as of December 2020, more than 70 million cases have been recorded and over 1.6 million deaths have been attributable to the virus [2]. Several studies have focused on individual determinants of Coronavirus 19 disease (COVID-19), such as age, sex, genetic polymorphisms [3], and underlying health conditions, whereas only a few studies have investigated the role of social determinants.

However, socio-demographic factors play a crucial role in shaping the pattern of COVID-19 positive cases and deaths across the globe. Some preliminary data would indicate that social determinants and existing health inequities could be linked to SARS-CoV-2 infection diffusion and COVID-19 severity [4,5]. Zhang and collaborators have shown that in the United States, population density, proportion of elderly residents, and poverty could be associated with higher incidence of COVID-19 [6]. Similarly, Fortaleza et al., in Brazil, showed a positive relationship between population density and the incidence of COVID-19 and between proximity to large cities and time-to-introduction and incidence rates of COVID-19 [7].

Unfortunately, a large proportion of studies evaluated social determinants at a national or regional level, whereas analyses carried out on microdata and at an individual-level could support a more accurate evaluation of the risk. Such data, when available, could provide essential information to support the government’s decision making body to strategically manage health emergencies in the present and in future if similar situations were to arise. In particular, since these conditions specifically put an individual at higher risk of being infected with SARS-CoV-2, populations that are considered to be more vulnerable according to these criteria could be given the resources needed to endure infectious outbreaks.

The aim of the present paper was to analyse the characteristics of the first 10,114 cases observed in Sicily, Italy and investigate, from an ecological point of view, some social determinants that could have had an impact on the SARS-CoV-2 spread at a census section level.

## 2. Materials and Methods

This study has been carried out by considering all SARS-CoV-2 cases observed among Sicilian residents between the 1 February 2020 and 15 October 2020. Sicily is the largest island in the Mediterranean Sea and one of the most populous (about five million inhabitants) of the 20 regions of Italy. The first case of SARS-CoV-2 in Sicily was observed on 24 February 2020.

In this study, SARS-CoV-2 patients have been considered eligible if they met the following inclusion criteria:–Being a resident of Sicily;–Having a laboratory-confirmed SARS-CoV-2 positive result of reverse transcriptase real-time polymerase chain reaction (rtReal-Time PCR) of nasal, pharyngeal, or nasopharyngeal swabs.

SARS-CoV-2 positivity in the previous months have been confirmed according to the notification status in the regional section of the national database collected and updated on a daily basis in the Sicilian region and provided by the Istituto Superiore di Sanità (ISS).

For each patient, the following information has been collected: age, sex, date of infection detection, residency, clinical outcomes (codified as Asymptomatic, Mild, Moderate, Severe, Critical, Deceased), and exposure route (categorized as Community/Nursing home, Home, Work, Pleasure places, Hospital, School, Trip and Unknown). The clinical outcome should be considered as the worst clinical situation experienced by each patient during the disease course.

Patients included in the analyses were anonymized and geocoding was performed by automated information system in accordance with a respect for privacy. Cases were thus attributed to section census and incidence rates were obtained for each small geographic area. Census section data were obtained by the last available census report that has been carried out by the Italian National Institute of Statistics (ISTAT) in 2011 [8]. Individual data were aggregated at the level of geographical units used for census purposes. A total of 34,064 census sections have been considered and these geographical areas are the smallest geographical units for which census data are available. For each census section, the following socio-demographic indices were considered: Total resident population, Residents/square kilometre, Male proportion (%), Residents aged <15 years (%), Residents aged 15–34 years (%), Residents aged 35–64 years (%), Residents aged >64 years (%), Residents with University degree (%), Residents with secondary school (%), Illiterate residents (%), Employed residents (%), Unemployed residents (%), Mobility extra urban (%), Home ownership (%), Home for rent (%), Buildings with production activities (%), Homes in bad conditions (%), Presence of immigrants (%), Presence > five homes per building (%), Number of residents per house (%), Home size per person (in square meter). SARS-CoV-2 incidence rates were calculated for each census section (*n* cases/total residents * 100,000) and census sections with SARS-CoV-2 rates higher than the regional value (202.3 cases per 100,000) have been considered at higher risk. The study was conducted according to the principles stated in the Declaration of Helsinki.

### Statistical Analyses

The normal distribution of continuous variables was assessed by Shapiro–Wilk test, and since all variables have been found to be non-normally distributed, they have been presented as median and interquartile range (IQR). Categorical variables have been summarized as absolute number (percentage). Incidence rates (number of SARS-CoV-2 per 100,000 residents per year) and 95% confidence intervals (95% CI) were calculated via Byar’s approximation.

Mann–Whitney rank sum test (for non parametric continuos variables) and chi-square test (for categorical data) were used to compare variables between census sections with SARS-CoV-2 rates higher than regional value vs. SARS-CoV-2 rates lower or equal to regional value. A backward stepwise multivariable logistic model was built to determine the association between sociodemographic variables and SARS-CoV-2 rates (0 when lower or equal to regional value and 1 when higher than regional value). The output of the regression model has been summarized as odds ratios (ORs) and 95% confidence intervals (95% CI). In the final regression model, missing variables have been deleted from the analyses.

All statistical tests were two-tailed, and statistical significance was defined as *p* ≤ 0.05. Analyses were performed using R Software analysis 3.6.1. R Foundation, Vienna, Austria [9].

## 3. Results

As of 15 October 2020, a total of 10,788 SARS-CoV-2 cases were observed in Sicily and 10,114 (93.8%—202.3 cases per 100,000; 95% CI = 198.4–206.2) were residents of Sicily and thus considered in the analyses. Table 1 summarizes the main characteristics of the SARS-CoV-2 infected population and highlights that September and October were months with higher incident cases (about 57% of the total).

For a large majority of patients (61.3%), the route of exposure was not well defined whereas home was the most frequent (15.95%) among those that have been reported. About 45% of all patients were asymptomatic and the fatality rate was 3.62%.

From Table 2, the worst clinical presentation was stratified according to sex and age group. Asymptomatic or mild cases were more frequent among young groups (about 50% up to patients aged 39 years) whereas severe/critical conditions showed higher prevalence among older patients (over 20% of patients aged 70 or more died or had severe or critical conditions).

The main investigated socio-demographic risk factors are reported in Table 3. Variables statistically significantly associated with higher rates were included in a multivariable logistic regression model. As reported in Table 4, overall, two variables (male prevalence and presence of immigrants) were found to significantly reduce the risk of SARS-CoV-2 cases (adj-OR = 0.99, 95% CI = 0.99–0.99 and adj-OR = 0.89, 95% CI = 0.86–0.92). The proportion of residents aged <15 years, residents with a university degree, residents with secondary level education, mobility extra urban, home for rent, and presence of more than five homes per building were found to increase the risk for SARS-CoV-2 incidence rates higher than the regional value. In particular, the prevalence of residents aged <15 years and extra urban mobility were characterized by the highest odds ratio (adj-OR = 1.10, 95% CI = 1.06–1.15 and adj-OR = 1.10, 95% CI = 1.07–1.14, respectively).

## 4. Discussion

The emergence of COVID-19 has changed not only the lives of a large part of the world population but also the classical approach of scientific research. The need to construct new knowledge about epidemiology, transmission, clinical characteristics, diagnostics, testing, and screening of this novel virus have required to us to share data in a timely manner. Unfortunately, in this context some issues have been neglected and there is a proportional paucity of studies on socio-demographic factors although some increasing evidence could support the strategic role of these determinants in reducing the diffusion of the outbreak. Our study aimed to investigate this topic and tried to understand the possible role of sociodemographic characteristics in determining SARS-CoV-2 cases in very small geographic areas during a low-incidence period.

Our main and most significant finding is that sociodemographic variables, which change geographically, could have a direct impact on the increase in COVID-19 incidence rates. In particular, we found that some factors seem to have a strongly statistically significant impact in reducing the risk of SARS-CoV-2 presence whereas others could increase this risk.

Among these latter variables, a particular role was observed for relative frequencies of residents aged < 15 years, home for rent in the census section, and extra-urban mobility. All three of these factors have been independently associated with an increase in the risk of about 10% per percentage increase in each variable. The reasons that could explain these strong associations seem to have both biological plausibility and consistency with some other studies.

As in our study, Ferreira et al. [10] observed higher SARS-CoV-2 incidence rates in areas with more children (12.9% vs. 11.1%; *p* < 0.001), although this association was not confirmed by multivariable analyses. However, several studies are suggesting that children could contribute to increasing the risk of SARS-CoV-2 transmission. In particular, children shed viral SARS-CoV-2 RNA (whether viable or not) in a similar manner to adults [11] but, with respect to the latter, they appear to be more frequently asymptomatic [12], exactly as in this study, and less frequently tested for SARS-CoV-2. All these considerations would support the possibility that children increase the transmission in settings with interactions like those in the household settings. According to this possibility, home was the most common exposure place for our patients.

A second important risk factor in this study was the relative presence of home for rent in a census section. In our opinion, and as reported by others [13], this could be an indirect socioeconomic deprivation indicator quantifying the magnitude of geographically determined social inequalities in health. For this reason, it should be interpretated according to two other variables that have a similar meaning as residents with university degree and residents with secondary level education.

As reported by others, these indices include material circumstances, the social environment, and psychological factors. Thus, these differences may not be primarily driven by income but have more to do with variations in social participation and ability to control life circumstances [14]. The increased risk observed in census section with higher prevalence of patients with higher education could also be explained by considering that these patients usually have prompt access to the health system, and thus to testing. Moreover, during the SARS-CoV-2 pandemic, low education level patients could have lost their work (or reduced their working activities) with a consequent decrease in their social contacts and the associated risk of infection. Finally, we have to consider that a large part of SARS-CoV-2 infections in Sicily during the first epidemic wave were attributable to trips in other Italian regions or other countries, and these latter exposures are typically due to working activities and are more frequent in patients with higher level education.

The contribution of the social contacts seems to be confirmed by the higher risk in census areas where a higher percentage of residents are involved in extra urban mobility for working activities. Overall, our results showed that some sociodemographic characteristics could affect environmental exposure to social contact with increased rates of COVID-19 infection. In this sense, immigrants could have been subjected to a reduced risk. During the lockdown, immigrants have been at higher risk of job loss or reduced work hours, and it cannot be excluded that they have also reduced their social contacts with the local population. All these conditions could have decreased their risk of getting the SARS-CoV-2 infection, whereas a further difficulty in accessing the healthcare system could have reduced their chances of being tested and notified.

A reduced risk of SARS-CoV-2 was also found in the census section with higher male prevalence. We are convinced that this finding could be related to the higher number of social contacts that women are used to having because of cultural roles and gender norms. In this sense, female gender has been associated with a higher likelihood of employment in essential services like health care and service industries [15] or greater likelihood of caregiving including childcare and teachers [16], which are services at higher risk of exposure to SARS-CoV-2 [17].

Finally, we have observed higher SARS-CoV-2 incidence rates in census sections with higher relative presence of more than five homes per building. This result could be easily explained by considering that densely populated areas could facilitate community spread of infectious diseases [18].

A major strength of our study is that it has included the total Sicilian population and, thus, our analysis does not suffer from potential selection bias. Moreover, we have considered a phase with low incidence during the which it is expected that the detection rate may be not affected by stress in healthcare facilities.

Despite of these strengths, we cannot exclude some important possible limitations. Firstly, sociodemographic data was obtained from a single decennial census of the population that was performed ten years (2011) before the present study. Thus, such classifications are questionable because of the changing composition over time of small geographical areas [19]. Secondly, exposure data are based on aggregated data instead of individual data. Moreover, we have not adjusted statistical analysis by smoothing techniques and spatial correlation. However, we are convinced that in an epidemic with a high number of cases and long observation period, estimates of disease patterns could be relatively stable. Finally, it should be considered that, as with all ecological studies, the design of the present study has to be considered at light of several limitations that could affect causal inference, including ecologic and cross-level bias, problems of confounder control, within-group misclassification, lack of adequate data, temporal ambiguity, collinearity, and migration across groups [20].

## 5. Conclusions

Despite the previously reported limitations, this is one of the first studies that evaluates the role of routinely collected socio-demographic data as a predictor of SARS-CoV-2 risk infection. The very high number of included cases (>10,000) and census areas (>34,000) makes our estimates of particular interest and suggests that socio-demographic data may play a role in mapping high risk micro-areas for current and future epidemic scenarios. Further analyses and studies could be required for corroborating these findings and increase their generalizability and accuracy during a period with higher incidence rates and more pronounced virus circulation.

## Figures and Tables

**Table 1 healthcare-09-00867-t001:** Characteristics of the SARS-CoV-2 infected population during the study period (Sicily; N = 10,114).

		N *	%
**Resident in Sicily**		10,114	100%
**Sex ***			
	**Males**	4704	46.51%
	**Females**	5410	53.49%
**Age (in years) ***			
	**0–14**	672	6.64%
	**15–24**	1396	13.80%
	**25–49**	3518	34.78%
	**50–64**	2357	23.30%
	**>64 to 99**	2097	20.73%
**Month of infection ***			
	**February**	5	0.0%
	**March**	1661	16.4%
	**April**	1005	9.9%
	**May**	189	1.9%
	**June**	52	0.5%
	**July**	244	2.4%
	**August**	1058	10.5%
	**September**	2699	26.7%
	**October**	3126	30.9%
**Exposure**			
	**Community/Nursing home**	494	4.88%
	**Home**	1613	15.95%
	**Work**	475	4.70%
	**Pleasure places**	431	4.26%
	**Hospital**	610	6.03%
	**School**	39	0.39%
	**Trip**	247	2.44%
	**Unknown**	6205	61.35%
**Worst clinical presentation**			
	**Asymptomatic**	4612	45.4%
	**Mild**	2655	26.25%
	**Moderate**	1469	14.52%
	**Severe**	457	4.52%
	**Critical**	163	1.61%
	**Deceased**	366	3.62%
	**Unknown**	392	3.88%

* Percentage sum may differ from 100% due to missing data.

**Table 2 healthcare-09-00867-t002:** Worst clinical presentation according to sex and age group.

	0–9	10–19	20–29	30–39	40–49	50–59	60–69	70–79	80–89	>89
**Females**										
Asymptomatic	63%	57%	49%	49%	49%	44%	41%	35%	20%	29%
Mild	24%	31%	34%	34%	30%	30%	26%	18%	16%	8%
Moderate	7%	7%	14%	12%	15%	16%	19%	22%	21%	10%
Severe	0%	0%	1%	1%	2%	4%	7%	10%	13%	17%
Critical	0%	0%	0%	0%	0%	2%	3%	3%	4%	3%
Deceased	0%	0%	0%	0%	0%	1%	2%	9%	22%	34%
**Males**										
Asymptomatic	55%	68%	61%	50%	45%	40%	37%	26%	22%	10%
Mild	28%	22%	25%	32%	30%	27%	21%	18%	11%	12%
Moderate	9%	4%	8%	12%	15%	20%	19%	21%	19%	17%
Severe	0%	0%	1%	1%	4%	6%	11%	13%	12%	7%
Critical	0%	0%	0%	1%	2%	3%	4%	5%	4%	2%
Deceased	1%	0%	0%	0%	0%	2%	4%	16%	29%	50%

**Table 3 healthcare-09-00867-t003:** Risk factors evaluated in census sections with SARS-CoV-2 infection rates higher or lower than those observed in Sicily during the study period (202.3 cases per 100,000).

	Lower Risk	Higher Risk	*p*-Value
**Total residents**	3,858,261	1,144,643	-
**Male proportion, N/Tot (%)**	1,866,982/3,858,261 (48.4%)	551,775/1,144,643 (48.2%)	<0.001
**Residents/Km2 (median, IQR)**	7.5 (1.4–16.2)	8.7 (3.4–16.3)	<0.001
**Residents aged <15 years, N/Tot (%)**	575,416/3,858,261 (14.9%)	171,968/1,144,643 (15%)	0.003
**Residents aged 15–34 years, N/Tot (%)**	960,244/3,858,261 (24.9%)	283,575/1,144,643 (24.8%).	0.013
**Residents aged 35–64 years, N/Tot (%)**	1,593,518/3,858,261 (41.3%)	474,983/1,144,643 (41.5%)	<0.001
**Residents aged >64 years, N/Tot (%)**	729,083/3,858,261 (18.9%)	214,117/1,144,643 (18.7%)	<0.001
**Residents with University degree, N/Tot (%)**	339,827/3,858,261 (8.8%)	107,254/1,144,643 (9.4%)	<0.001
**Residents with secondary education, N/Tot (%)**	985,145/3,858,261 (25.5%)	303,707/1,144,643 (26.5%)	<0.001
**Illiterate residents, N/Tot (%)**	72,671/3,858,261 (1.9%)	19,567/1,144,643 (1.7%)	<0.001
**Employed residents, N/Tot (%)**	1,142,995/1,462,432 (78.2%)	345,077/439,827 (78.5%)	<0.001
**Unemployed residents, N/Tot (%)**	168,874/1,462,432 (11.5%)	50,987/439,827 (11.6%)	0.41
**Extra urban mobility, N/Tot (%)**	387,388/3,858,261 (10%)	131,074/1,144,643 (11.5%)	<0.001
**Home ownership, N/Tot (%)**	1,061,798/1,515,512 (70.1%)	316,633/448,065 (70.7%)	0.033
**Home for rent, N/Tot (%)**	227,914/1,515,512 (15%)	68,696/448,065 (15.3%)	<0.001
**Buildings with production activities, N/Tot (%)**	128,935/1,372,383 (9.4%)	34,253/354,060 (9.7%)	<0.001
**Homes in bad conditions, N/Tot (%)**	35,558/1,372,383 (2.6%)	8015/354,060 (2.3%)	<0.001
**Presence of immigrants, N/Tot (%)**	99,618/3,858,261 (2.6%)	25,397/1,144,643 (2.2%)	<0.001
**Presence > five homes per building, N/Tot (%)**	83,538/1,372,383 (6.1%)	23,873/354,060 (6.7%)	<0.001
**Number of residents per house, median (IQR)**	2.49 (2.15–2.83)	2.5 (2.25–2.78)	<0.001
**Home size per person in square meters, median (IQR)**	38.93 (33.23–45.92)	38.83 (34.16–44.3)	0.31

**Table 4 healthcare-09-00867-t004:** Multivariable analysis on risk factors evaluated in census sections with SARS-CoV-2 infection rates higher than those observed in Sicily during the study period (202.3 cases per 100,000) (N = 298 observations have been deleted from the analyses due to missingness).

	Adj-OR	Lower 95% CI	Upper 95% CI	*p*-Value
**Sex proportion, (per % increase)**				
*Female*	REF			
*Male*	0.93	0.90	0.97	<0.0001
**Age group, (per % increase)**				
*Residents aged >14 years*	REF			
*Residents aged <15 years*	1.10	1.06	1.15	<0.0001
**Education level, (per % increase)**				
*Residents without University degree*	REF			
*Residents with University degree*	1.03	1.00	1.07	0.026
**Secondary school education (per % increase)**				
*Residents with primary or no education*	REF			
*Residents with secondary education or more*	1.05	1.03	1.08	<0.0001
**Extra urban mobility, (per % increase)**				
*No*	REF			
*Yes*	1.10	1.07	1.14	<0.0001
**Home for rent, (per % increase)**				
*No*	REF			
*Yes*	1.09	1.07	1.12	<0.0001
**Presence of immigrants (%)**				
*No*	REF			
*Yes*	0.89	0.86	0.92	<0.0001
**Presence of more than 5 homes per building (%)**				
*No*	REF			
*Yes*	1.06	1.05	1.07	<0.0001

## Data Availability

The data are available under reasonable request to the corresponding authors.

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
