# Peer review of "Socio-Demographic Factors Involved in a Low-Incidence Phase of SARS-CoV-2 Spread in Sicily, Italy"

_healthcare, 2021, doi:10.3390/healthcare9070867_

Round 1

Reviewer 1 Report

Abstract, authors must report 95% confidence intervals (CIs) for odds ratios as it is not appropriate to report estimate and p-value only. I would also suggest adding 95%CIs for other results such as rates.

Please use ‘patient’ instead of ‘subject’ as it is unethical.

Did authors consider any interaction between independent variable in the multivariable logistic regression analysis? Need a sentence to explain.

Please indicate in the analysis plan that what estimates from logistic regression are being presented (i.e., odds ratio and 95%CIs).

Please add 95%CIs for incidence rates presented and mention in the analysis plan.

How missing date were handled in the multivariable logistic variable? Please add couple of lines in the analysis plan.

Table 1, please remove “-“ sign in age and sex variables, write male and female rather than just M and F. add unit in bracket in age variable.

Table 2, please add sample size for each group presented in the column i.e., SARS-CoV-2 rates ≤ Regional value, and SARS-CoV-2 rates > Regional value. As most of parameters presented are measured as percent then how Mann-Whitney rank sum test was applied to compare these groups, and if chi-square test was used to compare these categorical variables, then there will be only one p-value for each categorical variable. Please add more clarity.

Table 3, reference category should be mentioned for each predictor presented in this table.

Reviewer 2 Report

The authors conduct a descriptive study and to investigate the social determinants that could have an impact in the virus spread among the 10114 SARS-CoV-2 Sicilian cases from 2020/201 to 2020/10/15.

Comments

I cannot understand the statistical analysis method in Table 2. Testing a categorical data is used the median (IQR) compared with groups, but it is not used the counts and percents.

1.Table 2:

Mann-Whitney rank sum test was used to compare non-parametric continuous variables between census sections with SARS-CoV-2 rates higher than Regional value vs.  SARS-CoV-2 rates lower or equal to Regional value, using a proportion(%) median (IQR).

In general, the categorical variables are analyzed by the Chi Square (χ2) test.

2.Next (Table 3), a multiple backward stepwise logistic analysis on risk factors evaluated in census sections with SARS-CoV-2 infection rates higher than those observed in Sicily during the study period, analyzing the continuous variables with a proportion(%) median (IQR).

3.As a question, the authors have recruited 10114 SARS-CoV-2 cases for this study (Table 1, count and percent).

4.In general, one way to summarize categorical data is to simply count, or tally up, the number of individuals that fall into each category. The number of individuals in any given category is called the frequency for that category.

5.Suggestions: the authors should be analyzed the 10114 SARS-CoV-2 cases, including count and percent for categorical variables, e.g., Chi Square (χ2) test in Table 2, and evaluated the associations between socio-demographic factors and infection rate category in Table 3 (categorical variables), or having other reasons.

6.In Table 2 for infection rates ≤ Regional value group, the residents aged <15 years is 13.86%; 15-34 years is 12.2%, 35-64 years is 53.2%; aged >64 years is 8.21%. Summarizing the age group, it should be nearing 100%, but it is 87.47%. In the infection rates >Regional value group, it is 14.36%, 12.28%, 53.5%, and 19.19% with add up equal to 99.33% as nearing 100%.

Round 2

Reviewer 2 Report

No further comment

Author Response

Thank you very much.